# The Potential of Selected Agri-Food Loss and Waste to Contribute to a Circular Economy: Applications in the Food, Cosmetic and Pharmaceutical Industries

**DOI:** 10.3390/molecules26020515

**Published:** 2021-01-19

**Authors:** Lady Laura Del Rio Osorio, Edwin Flórez-López, Carlos David Grande-Tovar

**Affiliations:** 1Programa de Ingeniería Agroindustrial, Facultad de Ingeniería, Universidad del Atlántico, Puerto Colombia 081008, Colombia; lldelrio@mail.uniatlantico.edu.co; 2Grupo de Investigación en Química y Biotecnología QUIBIO, Universidad Santiago de Cali, Calle 5 No 62-00, Cali 760035, Colombia; edwin.florez00@usc.edu.co; 3Grupo de Investigación en Fotoquímica y Fotobiología, Programa de Química, Facultad de Ciencias Básicas, Universidad del Atlántico, Puerto Colombia 081008, Colombia

**Keywords:** agri-food waste valorization, antioxidants, bioactive compounds, circular economy, colorants, fruit seeds, fruit peel

## Abstract

The food sector includes several large industries such as canned food, pasta, flour, frozen products, and beverages. Those industries transform agricultural raw materials into added-value products. The fruit and vegetable industry is the largest and fastest-growing segment of the world agricultural production market, which commercialize various products such as juices, jams, and dehydrated products, followed by the cereal industry products such as chocolate, beer, and vegetable oils are produced. Similarly, the root and tuber industry produces flours and starches essential for the daily diet due to their high carbohydrate content. However, the processing of these foods generates a large amount of waste several times improperly disposed of in landfills. Due to the increase in the world’s population, the indiscriminate use of natural resources generates waste and food supply limitations due to the scarcity of resources, increasing hunger worldwide. The circular economy offers various tools for raising awareness for the recovery of waste, one of the best alternatives to mitigate the excessive consumption of raw materials and reduce waste. The loss and waste of food as a raw material offers bioactive compounds, enzymes, and nutrients that add value to the food cosmetic and pharmaceutical industries. This paper systematically reviewed literature with different food loss and waste by-products as animal feed, cosmetic, and pharmaceutical products that strongly contribute to the paradigm shift to a circular economy. Additionally, this review compiles studies related to the integral recovery of by-products from the processing of fruits, vegetables, tubers, cereals, and legumes from the food industry, with the potential in SARS-CoV-2 disease and bacterial diseases treatment.

## 1. Introduction

The shortage of raw materials in food production worldwide has generated an environmental, economic, and social imbalance, causing the indiscriminate exploitation of natural resources, the rise in the cost of food products, and the daily increase in the number of people who suffer from extreme hunger [1]. According to the World Food Program (WFP), approximately 250 million people suffer from extreme hunger worldwide, falling behind the goal of reaching zero hunger by 2030 [2]. The current model of the economy (linear model) is based on the model inherited from the industrial revolution under the concept of the constant supply of products with a short useful life, forcing to produce more to satisfy the consumer’s constant needs. The linear economy (extract, manufacture, and disposal) increases the indiscriminate exploitation of limited natural resources that would give way to a significant environmental and economic crisis [3,4].

The by-products of the agri-food industry (peels, seeds, shells, pomace, and leaves) are useful due to their content of bioactive compounds (phenols, peptides, carotenoids, anthocyanins, and fatty acids), fibers, and enzymes of great interest for the production of functional foods and drugs against acute and chronic diseases, and antioxidants that can be applied to the cosmetic industry [5]. These by-products are those food losses and waste (FLW) that, according to FAO, refer to their decline in the successive food production supply chain for human consumption [6].

The generation of FLW worldwide is much higher in developed countries. According to Van der Werf and Gilliland [7], are produced 198.9 kg/year per capita of FLW in developed countries. In the United States, the FLW covers 40% of the whole food production chain [8]. North Africa and West and Central Asia account for 32% of the global FLW volume [9]. The European continent represents a third (20%) of the FLW generated worldwide [10], while in Latin America, FLW is estimated as 15% of total food production, which represents 6% of FLW worldwide [11]. Therefore, the generation and accumulation of FLW imply a significant impact on biodiversity, human health, and climate change [12]. For example, a 60% increase in greenhouse gas emissions [13] and malnutrition in the population are examples of negative impacts of FLW generation [14].

For the proper management of these by-products, it is necessary a decisive change for the agri-food system. The circular economy promises to be an efficient option in the medium and long term to prevent, reuse, or recover natural resources and derived by-products [15]. The goal is to re-introduce the by-products to the production line as raw material for obtaining new products with high health benefits and added value in the industry through sustainable technology to extract nutritional components [16].

On the other hand, the health crisis currently being experienced worldwide by the COVID-19 disease has impacted the food system within production processes [17]. Consequently, research has been carried out to generate alternatives for eliminating the virus and improving the food system to supply the world population [18,19]. For this reason, one of the objectives of this review is to compile various studies related to the integral recovery of by-products from the processing of fruits, vegetables, tubers, cereals, and legumes in the food industry, showing the potential of phenolic compounds present in their by-products in the SARS-CoV-2 virus and bacterial diseases treatments. Noteworthy is also the review of various works that obtained bioactive components, fibers, enzymes, and flours for their functional foods and the cosmetic industry’s application. This review is also intended to contribute to the paradigm-shifting towards a circular economy by highlighting various studies around the world that demonstrate the value and application of these wastes in other industries, reducing the indiscriminate disposal of waste.

## 2. Circular Economy

According to the demographic report, the world population stands at 7625 million people, with an annual growth rate of approximately 74 million people [20]. The United Nations (UN) has estimated that by 2050 the population will reach 9.2 billion people [20]. The current economic model (linear economy) has allowed rapid industrial and cultural development. However, its extracting, consuming, and disposal structure negatively impacts the availability of natural resources that supply the world population. Population growth also generates more natural resources and food consumption. Despite this, current resources are limited to supply the necessary food for the growing population and the generation of large amounts of waste [21]. These wastes are transported to sanitary landfills every year, reducing the available land for agriculture, causing damage to health and the environment [15]. In response to the problem, the use (or reuse) of waste would facilitate food supply to the growing population, reducing the indiscriminate use of human consumption resources [22]. The circular economy drastically transforms the culture and the system of extraction, production, and consumption towards a system of restoration and regeneration of natural resources’ value, limiting the excessive use of raw materials and energy and avoiding the unnecessary generation of waste [23]. In the circular economy, a system is established where the products are designed in such a way that the generation of waste is minimal or wholly eliminated, implementing a culture in which the product is designed to give it a second useful life, with added value and also, the reduction of energy and raw material consumption [24].

The transition towards a circular economy involves various social, economic, and environmental spheres, which generate opportunities for regeneration, renewal, and innovation in the agri-food industries, protecting the scarcity of resources [25]. The integration of higher income from circular activities and the minimization of manufacturing costs would affect demand, supply, and prices, generating indirect effects that accelerate the economy’s total growth and varying in favorable terms of GDP [26,27]. In the same way, high-quality recycling activities and skilled jobs in the transformation and remanufacturing of food losses and waste creating jobs through the development of local reverse logistics with SMEs generating net savings in raw material costs adopting a circular model [28].

In the environment, the circular economy directly influences the exploitation and deterioration of the ecosystem, reducing little by little to a null point, the excessive consumption of synthetic materials, such as fertilizers, pesticides, fuel, and non-renewable electricity that generate significant damage by polluting the air, soil, and water [29]. Likewise, circular economy generates a reduction of residues in the food chain by improving productivity and technology applied in the transformation and regeneration processes of natural materials, making responsible use of soil, aquatic resources, and the reduction of carbon dioxide emissions [30]. The circular economy’s social aspects have a drastic impact on the current culture of food consumption, with lower amounts of waste in homes and greater use of natural and energy resources [24]. In the same way, giving a second utility to packaging by the consumer [31,32]. The problem of social inequality caused by poverty and famine would decrease, obtaining more affordable food and greater availability of jobs due to the new circular technological systems and the creation of companies in conjunction with local industries, forming a fabric that will improve the quality of food products and consumer satisfaction [33,34].

The aspects that currently focus on the circular economy are guided by the Ellen MacArthur Foundation’s butterfly diagram, based on two elements capable of processing everything that surrounds us in such a way that nothing is thrown away or wasted [35]. These elements are divided into technical and biological cycles. What belongs to the biological cycle must be compostable (compost for the soil) or used to generate biogas (gas obtained from the decomposition of organic waste). On the other side, everything is recycled without losing quality through maintenance, remanufacturing, or reuse [36]. However, the diagram focused on a completely closed and consecutive cycle is hardly executable in the economy due to the increase in energy costs and unavoidable material losses [37]. Velenturf et al. [38] propose an integrated diagram in which the production-consumption system and the biophysical environment are related to the environment and the aspects determined in the circular economy. This diagram is based on the integration of avoidable waste and industrial materials within the production system. The uncontrolled biophysical environment relates to the human being’s production and consumption system, using the biophysical environment’s natural resources, turning them into industrial products. In this way, the biological (natural) and technological (industrial) terms are replaced. Raw materials can be of natural or industrial origin, forming part of biological and chemical processes for their reincorporation in the biophysical environment. Industrial materials are redesigned at the end of their useful life and do not affect the biophysical environment. Those that do not need to be redesigned are reintegrated, reducing uncontrolled leaks of waste by redesigning the production and consumption system’s infrastructure.

Incorporating this proposal in the food production processes would expand the circular economy concept because there are avoidable and unavoidable losses and waste in the food system. According to the Food and Agriculture Organization of the United Nations (FAO), waste can be classified as waste and losses. Losses occur during agricultural production and processing, while waste occurs during consumer consumption [92]. Therefore, food losses occur in the first four stages of processing, and the 5th stage is determined by consumer waste [39]. The integration of natural and industrial materials of Food Losses and Waste (FLW) is developed in a sustainable process where avoidable waste (food in the optimal condition that is not consumed) is not left aside, which can be returned to the environment (Figure 1).

Based on legal aspects, the analysis or treatment of food waste in a circular economy should focus on each country’s applicable laws [40,41]. Within the legal framework of the European Union (EU) in a global context of regulations and policies for the efficient use of resources and sustainable patterns of consumption and production, it plans to develop advanced routes of different recovery (incorporation into the food industries) to the usual processes (Animal feeding, composting and anaerobic digestion) [42,43]. The reuse and minimization of food waste will be dealt with, provided that these are suitable for human consumption and do not generate toxicity during their treatment [44].

The Treaty on the Functioning of the European Union contemplates the international transport of waste, which under this regulation waste is classified to determine its valuation and reuse subsequently. Agri-food waste (such as peels, seeds, vegetable, and cereal residues) are classified as “green list,” which are determined as non-infectious, and it is planned to incorporate them into the treatment process [45,46]. In the transition to a circular economy, the European Parliament initiated the regulation of the product’s life cycle in its entirety, from primary production to waste in conjunction with the management and market of secondary raw materials (food by-products) [47,48]. Countries such as Germany, France, and Italy, in the face of established regulations, have promoted government initiatives for the use of food waste that are not suitable for human consumption in the production of feed and composting [49].

In Japan, the Food Recycling Law was established in 2001 [50], which establishes that companies and industries in the food chain (from agricultural production to consumption) are participants in reusing waste and reducing food waste. This law encouraged companies such as Eco-Farm to use plant residues from food industries to make organic fertilizers that are used in crops that supply the raw material in food production [51]. In countries such as Korea, Taiwan, and Thailand, food waste is promoted and regulated by-laws to feed for ruminants, pigs, and poultry [52].

## 3. Popular Technologies for the Re-Use of Food Loss and Waste from the Agri-Food Sector

Food waste currently generates the main problem for natural resources, causing shortages, instability in the environment, and damage to public health through their poor disposal [53]. For many years the solution has been to dispose of them in sanitary landfills or otherwise burn them to reduce the amount of waste accumulated in these lands [54]. Due to this problem, some countries worldwide have developed and implemented technologies to generate the most effective use and recovery of food waste (Table 1), avoiding its disposal to landfills and reducing this waste [55]. In Europe, there is a broad practice of treatment of food waste through composting as the anaerobic digestion, due to European legislation that restricts the final disposal of the organic fraction without prior processing (Directive 1999/31/EC of 26 April 1999) [56,57].

The leading countries are Germany, Spain, and Switzerland [58]. In the European Union, approximately 30 million tons of separately collected food waste are composted or digested annually in almost 3500 treatment plants [59]. Pruning waste accounts for more than 50% of food waste, which is processed in more than 2000 composting plants, which is the system that dominates over anaerobic digestion, resulting in 90% of the treatment of food waste [60]. Compost production is relatively easy and inexpensive to implement on a local, national and regional scale, and it can be accompanied by biogas production, increasing the economic value generated per treated ton of food waste [61]. Additionally, biogas production for energy purposes through anaerobic digestion is considered an energy recovery technique from food waste [62].

## 4. Agri-Food Industry

The agri-food industry is structured by the primary sector dedicated to agriculture, fishing, livestock, forestry, and the agri-business, where raw materials’ processes and transformations (from the primary sector) into semi-finished products occurs for human consumption [70]. Similarly, this industry is subdivided according to the primary sector’s different activities and the raw material’s processing. Within processing, there are the dairy industry, the fruit and vegetable industry, the milling industry, the meat industry, the fish and seafood industry, the oil and fat industry, and the beverage industry, including alcoholic beverages [71]. Over time, commerce and the agri-food industry have been characterized by technological innovations implementing efficient production processes, pursuing cost reduction and competitiveness in the global market. The main goal is to satisfy the needs of the different population segments through technical and efficiency improvements [72].

The agricultural sector’s production processes in the agri-food industry involve transforming the raw material and subsequent distribution to the consumer (Figure 2). These stages are developed from the cultivation phase, where sowing systems are carried out according to the type of raw material to be cultivated, intensively (higher production in little space) and extensively (in greater surface area) [73]. Subsequently, harvesting (depending upon the fruit, the goal product, maturation stage, among others) is maintained at temperatures and humidity required to preserve the useful life in the distribution and consumption stages [74]. More than 5000 million metric tons of agricultural raw material are processed, obtaining millions of products globally [75].

On the other hand, the high number of products generated in the market, the lack of technological innovations in the processing of raw materials and infrastructure, inadequate food safety protocols, overcooking, unwanted sizes and weights in food, the defects of the packaging and labeling, are aspects that contribute to the food waste generation during the food supply chain, also with bad cultural habits for the purchases and consumption [77]. A large amount of waste generated worldwide is a big concern for governments and policymakers because of the immense contribution to environmental, economic, and social problems [78]. These wastes are generally unproperly used. On the contrary, they are discarded to landfills that increasingly occupy more land, contaminating soils, surface, and underground waters, increasing the ecosystem’s imbalance due to the agricultural sector’s indiscriminate production [79]. For this reason, it is necessary to study their chemical composition and their potential to introduce them as a raw material in several industrial processes, evolving towards a circular economy [78]. Following the above, this section will discuss the different agricultural industries such as the fruit and vegetable industry, the tuber industry, and the cereal and legume industry, establishing the production data, the operational processes where the residues are generated, and their main components.

### 4.1. Fruit and Vegetable Production

One of the industries with the highest production globally is horticulture, covering approximately 38% of agricultural production and 65% of vegetable production in the agri-food industry [80]. Among the primary fruit and vegetable producer countries are China, India, Turkey, and the United States (Figure 3). According to the Food and Agriculture Organization of the United Nations (FAO), during 2018, the production of fruits and vegetables worldwide exceeded 1800 million tons, increasing by 1.53% compared to the year 2017, and it is estimated that for the period 2020–2021 a considerable percentage will decrease due to the pandemic crisis of the COVID-19 disease that affects agri-food production globally [81,82].

Among the fruits and vegetables with the highest production are tomato (*Solanum lycopersicum*), apple (*Malus domestica*), onion (*Allium cepa*), orange (*Citrus × sinensis*), Cole (*Brassica oleracea* var. Capitata), gherkin (*Cucumis sativus* L.), eggplant (*Solanum melongena*), mango (*Mangifera indica*), carrot (*Daucus carota*), guava (*Psidium guajava* L.), and pepper (*Capsicum anuum*), which are processed for the production of value-added foods in the food industry (Figure 4) [80].

Within the transformation of fruits and vegetables, specific industrialization processes are developed such as dehulling or peeling, blanching, core removal, and pulping, which have temperature, humidity, and pH control systems for optimal conservation (Figure 5) [83]. These processes vary according to the type of fruit or vegetable and the final product obtained. However, it is possible to identify some basic transversal processes in the different industrialization processes, such as cleaning, sterilization, and pasteurization, that determine food safety [84]. The processing of fruits and vegetables is carried out to extend the shelf life and generate various food consumption options [85]. For example, various products with high-value in the market, such as preservatives, juices, jams, pasta, concentrates, and dehydrated products [86,87]. However, the horticultural industry generates significant losses of optimal raw material for human and animal consumption due to its high production. The food loss and waste correspond to biomass rich in bioactive compounds, enzymes, vitamins, and fibers, obtained through extraction processes and chemical transformation for animal and human nutrition or even to be used in biorefinery applications [88].

#### 4.1.1. Food Loss and Waste (FLW) Products from the Fruit and Vegetable Industrial Processing

The processing of fruits and vegetable generate several FLW such as peel, skin, seeds, stem, pomace, and bagasse [91]. These by-products generally come from specific processes such as pulping, peeling, straining, and blanching, necessary for food processing and conditioning [86]. However, those food losses and waste products (FLW) are not used in the production process. On the contrary, they are classified as waste by dumping them in sanitary landfills or leaving them in the field, increasing the deterioration of the environment and scarcity of resources and losing the opportunity to contribute to the feeding of a growing world population that with the current production of resources will not be enough [6].

According to the Food and Agriculture Organization of the United Nations (FAO), food loss and wastes in the fruit and vegetable industries amount to 60% of the total production [88], where losses represent between 3 and 50% of the total of processed raw material (Table 2). In avoiding the excessive use of natural resources, the use and valorization of these by-products are of great importance both for the environment and for the population’s supply, preparing nutritional foods with health benefits, contributing to the transition from a linear to a circular economy, also known as a “bioeconomy” [91,92].

#### 4.1.2. Peels

Fruit and vegetable peels are generated in the peeling stage, where the pulp is separated from the raw material covering (Table 3). This operation is carried out mechanically or manually, depending on the production and technology used [101]. Currently, 500 million tons of peel residues are produced in the fruit and vegetable industry, covering from 3 to 50% of the fruit or vegetable’s total fresh weight [102,103]. Orange (*Citrus sinensis*) is the fruit with the highest processing, generating 16 million tons of waste every year [103]. In general, fruit peels present a good source of cellulose, hemicellulose, and phenolic compounds with the potential to be used as raw material to generate new products [102]. The uses could include preparing animal feed organic fertilizers, cosmetic, and pharmaceutical applications due to their bioactive components such as phenols, peptides, terpenes, providing antioxidant, therapeutic, and nutritional properties [91].

#### 4.1.3. Seeds

The seed is a raw material fraction that is mostly wasted in processing fruits and vegetables. Also, they constitute between 15% and 40% of the total fruit (fresh weight). This by-product is generated in the pulping stage, where only the pulp is extracted [109]. Fruit seeds contain many nutrients such as proteins and lipids in 35% and 25%, respectively (Table 4). Tomato seed, one of the most abundant residues due to its extensive production in the market for pasta and sauce preparation, provides an essential source of unsaturated fatty acids, especially linoleic acid. Likewise, lycopene is the carotenoid responsible for the fruit’s red color with greater abundance in the peel [95,110]. On the other hand, the mango seed is a promising source of edible oil extraction and has been of great interest for its fatty acid profile similar to cocoa, butter, and as a source of phenolic compounds and phospholipids [111].

#### 4.1.4. Pomace and Core Fruit

The pomace represents the mixture between seed residues, skin, fiber, and pulp of the fruit or vegetable, generated in large quantities in the straining and pulping stage [121]. The pomace is characterized by containing a significant amount of fiber (between 40 and 60%) and bioactive compounds (Table 5) [122]. For example, grape pomace obtained from juices and wine production represents 20% of the total processed grapes [123], providing various phenolic constituents such as catechins, anthocyanins, flavanol glycosides, phenolic acids, and stilbenes [98]. On the other hand, the pineapple core (an inevitable residue in its processing) is an essential fiber source [93]. However, it is an edible residue that can be used as a dietary source.

Additionally, it contains an enzyme called bromelain with anti-inflammatory properties. It provides a more remarkable ability to digest food in the body [124]. Also, in the food industry, it is used as a meat tenderizer, in the processing of fish and marine products such as the production of oyster sauce, in the manufacture of cookies (to eliminate gluten), as a substitute for sulphones used to prevent browning of fruit juices, in white wine and beer clarification [125]. In the same way, it is used in the pharmaceutical industry to treat dyspepsia, infections, cellulitis, edema, and cancer [126,127].

### 4.2. Production of the Root and Tuber Industry

The food products with the highest proportion of carbohydrates are found in roots and tubers derivatives such as potato (*Solanum tuberosum*), yam (*Dioscorea esculenta*), and cassava (*Manihot esculenta*). They are considered essential foods due to their healthy nutrients content with an antioxidant, antimicrobial, hypoglycemic, and immunomodulatory capacity based on bioactive compounds such as phenols, saponins, bioactive proteins, and phytic acids [134]. This industry has an annual production of 820 million tons, mainly consumed in developing countries due to its low cost and caloric input [81]. Africa’s consumption is approximately 120 million tons, constituting 15% of world production, followed by the Asian continent and Latin America, with a consumption of 13 and 5%, respectively [135]. On the other hand, 40% of the production is industrialized for elaborating products with added value in the market [135], corresponding to flours, starches, fresh and processed products, with a demand of more than USD 92.12 billion [136,137,138]. These foods are used both for fresh consumption and as a raw material in the industry for the production of additives (10%), frozen products (40%), snacks (10%), bakery (25%), and pastries (15%) [136,139].

Among the leading world producers of tubers is China, with 39% of the total production, followed by Nigeria (30%), India (14%), Thailand (9%), and the Congo (8%) (Figure 6). Of the world production of roots and tubers, approximately 332 million tons of raw material are processed in the food industry [134]. Traditionally, tubers are marketed as fresh products, but due to the growing demand for production, new products have been developed, expanding their application forms [140]. Within the variety of most processed tubers are potato (*Solanum tuberosum*), cassava (*Manihot esculenta*), sweet potato (*Ipomoea batatas*), and yam (*Dioscorea alata*) [134], developing products as a functional ingredient in the industries of bread, flour, and fermented beverages such as beer [141].

The tubers’ transformation process varies according to the target product, use, variety, and tuber type (Figure 7) [142]. Generally, the most common processes are characterized by simple physical methods for maintaining the fresh product’s organoleptic properties [143]. The food conditioning has staged peeling, chopping, and reducing the particle size to produce starch or flours. These flours are used both in the preparation of homemade foods such as tortillas, arepas, fritters, cakes, among others, and in the industry of products for direct consumption such as bread, desserts, and instant soups [143]. Starch is the main cassava processing product used as raw material and ingredient for producing food-processed products such as bread, instant soups, rice, frozen products, and processed fruits [144]. It is estimated that starch production will increase annually, up to 376,000 tons globally, for its commercialization during 2019–2024 [145].

#### 4.2.1. Food Loss and Waste (FLW) Products from the Root and Tuber Industry

An estimated 320 million tons of tubers are globally processed every year, a second-place after food grains processing, required as an essential energy source [134]. The category of waste should be considered in context within the food chain, differentiating between waste and losses, concepts discussed in Section 2. These processing losses generate many by-products (approximately from 5 to 30% of the total raw material), such as the peel, bagasse, and pulp fractions in stages of size reduction (Table 6). These losses are significant biomass for its application in different industries due to its bioactive components and nutrients in producing new products [4].

#### 4.2.2. Peels

The processed root shells are between 2 and 15% of their total weight, usually generated in the peeling stage [143]. More than 11 million tons of peels are currently generated worldwide, where a small percentage are processed and used as feed animal ingredients due to the protein content. However, they are harmfully disposed of in landfills and water, mishandling these wastes [154]. These residues are essential due to their lignocellulosic characteristics (Table 7), which may vary according to their origin and the region where they are cultivated [155]. However, cassava peel contains linamarin and lotaustraline, toxic components found in a higher proportion in bitter cassava peels, the main variety processed in cassava starch manufacturing [156]. This component can be removed by heat treatments with an inactivation temperature of 75 °C [157].

#### 4.2.3. Cassava Bagasse

Bagasse is generated during the sieving or straining stage, where the pulp is separated from the fibrous material, constituting 10% of the processed cassava residue [150]. This residue contains a high percentage of water and starch residues because it is soaked in water in the starch separation stage. Under these conditions, bagasse presents a higher volume than the raw material represented by 60 to 90% of water and 10 to 30% of starch and fiber in dry weight [163]. Due to its fiber content, it provides lignocellulosic characteristics due to the cellulose (22%), hemicellulose (35%), and lignin contents (Table 8) [164].

### 4.3. Production of the Cereal and Legume Industry

Cereals are the leading staple food that provides the body with fuel and energy, thanks to the fact that they provide carbohydrates (between 50 and 70%), proteins (between 30 and 35%), lipids (2%), and vitamins (20%) [166]. On the other hand, legumes are an essential part of the diet due to their high protein content (up to 57%), carbohydrates (between 40 and 50%), fibers, and fats (between 8 and 15%) [167] generating other consumption alternatives in addition to meat for human diet [168]. Cereals such as corn (*Zea mays* L.), wheat (*Triticum aestivum*), rice (*Oryza sativa*), cocoa (*Theobroma cacao* L.), sorghum (*Sorghum vulgare*), oats (*Avena sativa*), rye (*Secale cereale* L.), and barley (*Hordeum vulgare*) [169,170,171] have high economic importance supplying a large part of European, Asian, and American countries with a consumption of approximately 2500 million tons [172]. They are currently relevant, developing new processes to innovate commercially in different aspects [173]. Among the legumes with the most significant growth in production processes are beans (*Phaseolus vulgaris*), chickpeas (*Cicer arietinum*), peas (*Pisum sativum*), lentils (*Lens culinaris*), and pigeon peas (*Cajanus cajan*) [174,175], with a variety of precooked canned products, powders such as condiments and additives, instant soups, among others [176]. The production of cereals and legumes is led by China and India, respectively (Figure 8), being the foods with the highest processing after fruits and vegetables. According to statistics from the FAO, cereals’ production constituted 73% and legumes 26.8% of this industry’s total in 2018 [81].

In the industrial processing of cereals and legumes, food products such as flour, corn flakes and oats, and canned legumes are prepared, while edible oil is extracted from their seeds, such as soybean oil [177]. The cereal grains and legumes must go through specific stages for their conditioning according to the final product’s physical characteristics, such as dehulling, grinding and sieving, and mechanical or solvent extraction of vegetable oils (Figure 9) [178]. These stages generate residues such as shells, cakes, or pellets from the oil extraction process and a fraction of raw material during the milling process [179]. The losses are used mostly for the production of energy and animal feed [180]. However, they have lignocellulosic characteristics and significant protein content that broaden their field of application to the food industry, such as the production of flours to prepare bread, cakes, and soups [179].

Cocoa is cultivated mainly in Europe, Asia, Africa, and America, with Africa as the largest producer (68% of the total world production), followed by the American continent, representing 15% of the world production [183]. The cocoa powder’s industrial processing generates various products such as cocoa butter, cocoa liquor, and cocoa powder, making sweets and chocolate (with the highest production) [184]. The European continent is the largest chocolate producer with sales of 47% in the world market, followed by the United States, with 20% in sales [185].

In chocolate production, the cocoa is first conditioned (fermented and dried) before transportation to the production plant [184]. Within the processing of chocolate, there are stages (Figure 10) such as the roasting of the grain, where the primary purpose is to highlight the flavor, continuing with the shelling, where the shell is removed, then the seed enters the milling stage, for subsequent pressing and extraction of cocoa butter, which is finally conditioned for chocolate preparation [132]. The residues generated are found in the critical stage (cocoa cake or powder) and the deshelling (shell). The cocoa cake is used to make sweets and confectionery, while it has been used for animal feed to a lesser extent. Moreover, the shells are used to a lesser extent in animal feed, and in most cases, they are determined as waste taken to landfills [186].

The largest barley producers include the European Union, Russia, Canada, Ukraine, Australia, and the United States [188]. For the 2019–2020 crop year, the United States Department of Agriculture (USDA) estimated a global barley area of about 51.6 million hectares [189]. On average, 35% of barley is used as animal feed and 65% in beer manufacture [188]. In 2019, world beer production amounted to around 1.9 billion hectoliters, most of which were produced in China [189]. The barley grain undergoes processing for beer preparation, firstly malted, and then passes to the maceration and filtration for alcoholic fermentation. Finally, the maturation steps allow obtaining a high-quality beer (Figure 11) [190]. In the filtering stage, the malt bagasse is generated from processing as waste, and only a small fraction is used as animal feed [191]. In general, 20 to 21 kilos of brewing barley are needed to produce 100 L of beer. That means that from 100 L of beer produced, 5 to 6 kg of bagasse would be generated (20 to 25% of the barley’s total weight). On a global scale, 1900 million hectoliters would generate 475 million bagasse bags, an alarming figure mainly limited to animal feed production [192].

#### 4.3.1. Food Losses and Waste (FLW) Generated from the Cereal and Legume Industry

China is the world’s largest cereal producer, followed by the United States, India, and Indonesia, with high consumption of processed foods such as croquettes, flours, and snacks [193]. However, they are also the main generators of FLW, especially in China and the United States [194]. In the processing of cereals and legumes, specific processes are required for optimal transformation, such as size reduction (if the final products are flours and canned goods), extraction of shells and functional compounds, pressing, drying, among others [195]. These stages also generate enormous losses (Table 9), with variations according to the raw material and the target product. Grain and legume shells, cake (cocoa or cereal), bagasse (beer), and pulp waste are among the essential wastes generated, which are generally discarded in landfills or water sources, increasing the pollution of the environment and wasting their chemical and nutritional potential [76].

#### 4.3.2. Husks

Husks residues are generated in the early stages of grain conditioning before the transformation in the dehulling and milling phase, where the coating is removed, and the endosperm is exposed [173]. In the wet milling process, it is possible to produce starch, sugar, syrup, or oil for human consumption from corn and, to a lesser extent, from sorghum [182]. Therefore, many of these cereals generate husk residues in large quantities, one of the most productive industries in the agri-food industry, such as chocolate, vegetable oils, and flours [111,173,189]. Among cereal husks’ properties, the high composition of cellulose and hemicellulose are highlighted with a higher proportion in the husks of beans, cocoa, and corn (Table 10).

#### 4.3.3. Malt Bagasse

The brewing industry generates bagasse during the pressing and filtering of the must due to barley’s saccharification. It is a by-product rich in fiber (60–70%) [190], carbohydrates, protein (between 20 and 60% on a dry basis, (Table 11), and a lignin content (4%) [211]. The wet bagasse is susceptible to microbial contamination, being a compromising situation for its use in the food industry [190]. Different preservation methods should be used for moisture removals, such as freezing, drying, and preservatives that reduce moisture by up to 90% [212]. Because bagasse is a waste with high fiber content, it is viable for its use in confectionery, pastry, and other food products and its application to create biorefineries and animal feed [192,213].

#### 4.3.4. Waste Cake

The edible vegetable oil industry generates solid waste, such as waste cakes, produced during the extraction process, either by pressing, with solvents or by centrifugation [190]. They are called cakes since, by pressing, a crushed bread shape is obtained. However, they are also presented in coarse flours or pellets [214]. These residues are generally used in animal feed due to their high protein content required in the animal diet [215]. The cake generated by soybean oil production contains a high percentage of proteins (Table 12). The discard cake is highly exported to the United States, Argentina, United Kingdom, and Mexico for animal feed and interest for its application in the food industry (for the production of flour, sausages, cereals, and tofu) [216].

## 5. Potential of Food Losses and Waste as a Raw Material in the Industry

### 5.1. Food Industry

One of the significant waste producer industries is the food industry, accounting for 70% of agricultural waste in the world [217]. Due to this problem, the generation of new waste recovery mechanisms is of great interest to the care of resources and the environment [218]. In the search to introduce these residues to food production, analyzes have been carried out against the components that can be exploited within the residual biomass [219]. The term by-product implies reusing waste as raw material, source, or ingredient, integrating it into the production process, turning a cost into a benefit [22]. Likewise, to reduce waste, action must be taken from the source, reducing the excessive amount of raw material used in industries and implementing monocultures through the formulation and execution of comprehensive management plans [220]. That is why the investigations of extraction, characterization, and application of residues from the food industry are of great interest for their use as by-products, giving alternatives on the excellent management of residues and the minimization of the indiscriminate use of natural resources contributing to a transition from the linear economy to a circular economy [221].

#### 5.1.1. Flours

Flours are physically characterized as fine powders obtained from the cereal grinding or starch present in food. Among the most common are wheat, corn, sorghum, rye, and oat flour. It is considered one of the main foods in the human diet [222]. In recent years, the world’s flour production has been characterized by its fortification by adding compounds and vitamins, supplementing the required daily nutritional balance [223]. The seeds, peels, and fruit pomace generated in the processing of food products present nutritional and sensory components, which are of great interest for preparing functional foods, thus reducing the addition of components in flours to obtain food products [224]. Consequently, studies have been carried out for the partial substitution of wheat, oat, and cornflour for peel, seed, and fruit pomace flour to prepare functional foods (Table 13), which have shown a favorable impact on the products obtained, increasing the nutritional content and antioxidant activity (due to the increase in phenolic content), without altering the physical and organoleptic properties.

#### 5.1.2. Colorants

The food colorants market is currently valued at USD 5 billion in 2020 and is estimated to grow at USD 6.8 billion by 2025, with a compound growth rate (CAGR) of 5.4% [236], where the largest market is in the European continent, followed by Asia and North America [237]. The industry of colorants is classified into two categories according to their origin. Synthetic colorants are obtained from petroleum derivatives [238], and natural colorants, obtained from fruit extracts and vegetables, can provide antioxidant capacities and increase nutritional food value [239]. Natural colorants are considered safer for food than synthetic colorants because of consumers’ health and environmental concerns [240]. According to a study conducted by the Clean Label Alliance, 75% of consumers are willing to pay a high price for clean label products (minimally processed products with the least amount of synthetic additives) [236]. However, Food safety would be put at risk by allocating raw material for food for industrial use [241].

Among the natural colorants available in food by-products are betanin, lutein, riboflavin, and anthocyanins, mainly found in fruit peels’ residues [238]. Studies have been carried out at laboratory scale and very few at pilot scale (Table 14) to use residues to extract colorants and their application in the food industry as additives and as a source of nutrients.

#### 5.1.3. Enzymes

Enzymes are of great importance in the industrial sector, as they allow sustainable development by reducing the chemical load of processes, eliminating toxic substances, and reducing pollutants [252]. Currently, the industrial enzymes market encompasses USD 5.9 billion, with a USD 8.7 billion projection in 2026 and a CAGR of 6.5% [253]. Among the most popular enzymes in the industry are lipases, carbohydrases, proteases, and polymerases applied in beverages and processed meat, dairy, fruits, and vegetables [253]. Among the leading countries in the enzyme industry market is the United States, China, the United Kingdom, Australia, Brazil, and Argentina [254]. The food industry produces large amounts of processing waste annually, most of which are lignocellulosic, containing a wide variety of enzymes considered critical raw materials in production and processes [255]. For example, bromelain extracted from pineapple can improve food digestion and soften beef to transform into value-added products in the market [256]. Lipases have various applications as food additives in the modification of taste, the synthesis of esters with an enjoyable antioxidant activity, hydrolysis of fats to produce detergents, wastewater treatment, and lipids removal in the cosmetic and pharmaceutical industry [257]. In this way, studies have been carried out for the enzyme extraction from agri-food waste (Table 15).

### 5.2. Cosmetic Industry

Cosmetics comprise a large industry for the care, protection, and improvement of the skin. This industry generates more than 500,000 million dollars, and it is estimated that by 2023 it will increase by 57% due to the large consumption of these products by new generations [267,268]. Among the countries with the highest cosmetics consumption, the United States is the leader with approximately 7000 million dollars, followed by Japan, Russia, and the United Kingdom [95]. Cosmetics are classified according to their functionality as hygienic (deodorants, cleansing foams, soaps), decorative (hair dyes, eye, and face makeup), corrective (lighteners, epilators, depigmented), and protective (sunscreen, moisturizers, lubricants) [268]. Each product is composed of a base substance, an active component, and the raw material or main ingredient used. Therefore, for the formulation of cosmetics, chemical compounds such as parabens and formaldehyde are used as preservatives. However, there is some uncertainty in using these components to affect metabolism, inducing a public concern for their massive application [269]. The use of these compounds has been reduced, incorporating cosmetic products with natural ingredients of fruits and cereals such as rice, orange, and oat bran. The process includes extracting their bioactive compounds and applying them in formulations as moisturizers and antioxidants for skincare [270].

#### Antioxidants

Antioxidants scavenge free radicals and prevent oxidation in body cells [271]. Cosmetics are mainly composed of antioxidants for their properties to reduce cell aging, applied in creams, powders, and oils for the skin [272]. One of the antioxidants’ properties with the most significant use on the market is the increase in skin protection against UV rays when added to creams with sunscreen, preventing the destruction of collagen and photoaging [273]. Antioxidants such as phenolic compounds, also defined as polyphenols, are derived from plants’ secondary metabolism [274]. They are commonly found in foods such as vegetables, fruits, and legumes consumed daily by humans, present in the pulp and seeds, skins, and pomace [39]. For instance, the extraction and application of phenolic compounds from the loss and waste of food have been of great interest due to their antioxidant capacity and to avoid the massive use of food to preserve food safety worldwide. [267]. For that reason, seeds and peels are considered antioxidant sources and are incorporated in the formulation of sunscreens (Table 16).

### 5.3. Pharmaceutical Industry

There has been a considerable increase in the manufacture of synthetic antimicrobial compounds against microorganisms that cause respiratory diseases, infections, or conditions in humans [282]. Worldwide, the pharmaceutical sector reached a CAGR of 8.8%, exceeding $1 billion in 2017 [283]. According to a study carried out by IQVIA, the sector should experience market growth between 3 to 6% over the next five years, exceeding 1.5 trillion dollars in 2023 [284]. However, the synthetic compounds in pharmaceutical products contain contraindications and side effects that can lead to the appearance of new health alterations and create resistance in bacterial strains [285]. Kardan et al. [286] evaluated the gene expression of the efflux pump (a mechanism that evaluates the resistance of microorganisms in drugs) in 31 clinical isolates of strains of *Mycobacterium tuberculosis*, determining gene mutations in 66.6% of the isolates (overexpression of the pump flow). Due to this, medicinal research against new alternatives in the preparation of drugs that do not affect health in the background has been of great interest in recent years, emphasizing bioactive compounds from plants, fruits, and functional foods [287]. The food industry generates many by-products worldwide that are discarded from the food transformation process that have a high content of compounds required as a raw material in the manufacture of medicines [288]. For this reason, research on the use of citrus fruit by-products has had a significant boom in recent years due to its antiviral, anti-inflammatory, antibacterial, and antifungal properties.

The bioactive compounds present are useful to produce medicines applied in the treatment of acute and chronic diseases. These residues are usually sources of phytochemical compounds and essential nutrients for developing and recovering the human body [289,290]. The extraction of these compounds from by-products is carried out by several routes [291]. In the case of phenolic extracts such as hesperidin, nobiletin, and tangeretin, there are currently ecological techniques such as hydroalcoholic extraction [292], which is 90% efficient both on a laboratory scale and on a pilot scale, and hydrodistillation applied mostly to the shells of citrus which provides 17% more performance than conventional methods [292,293]. In the same way, controlled hydrodynamic cavitation is the route with the highest performance overcoming acoustic cavitation to pilot scale for most applications, including natural products extraction [294,295]. Ultrasonic extraction with organic solvents offers advantages with much faster extraction rates than the medium and low energy consumption in conjunction with the MTT assay method that reflects the cellular viability due to the antimicrobial capacity of bioactive compounds [296].

#### 5.3.1. Antibacterial and Anticancer

Agri-food waste can increase the bioavailability of a large number of drugs as they are an excellent source of nutrients and phytochemical compounds capable of contributing to a healthy diet [297]. They are a good source of organic acids, sugars, and phenolic compounds such as flavonoids and anthocyanins, which have therapeutic applications due to their antibacterial, antifungal, anti-inflammatory, immunomodulatory activities under various conditions and an antioxidant potential against free radicals [298].

From the point of view of the recovery of residues such as peels, pomace, and seeds, commonly generated by the industries producing juices and other food products [231], the various investigations carried out on the bioactive compounds in these residues (Table 17) have been of great importance.

#### 5.3.2. Antivirals

According to biological terminology, the word virus originates from the Latin *virus* meaning toxin or poison [306]. Viruses are acellular infectious agent particles with nucleic acids (deoxyribonucleic acid, DNA, and ribonucleic acid, RNA) [307]. They are classified mainly by their phenotypic characteristics such as nucleic acid type, proteins, replicative cycle, and the viral infection produced [306]. A protein envelope generally encapsulates some called a capsid, others by a cell membrane or an envelope derived from the infected cell [308]. The evolution of viruses has increased their ability to multiply within the infected cell because they cannot do it by themselves due to the lack of necessary molecular material [309]. Viruses have been the leading cause of severe and chronic infections, causing the death or degeneration of host cells, capable of multiplying rapidly in the human body. Due to their rapid evolution, several mutations have produced new deadly diseases [308]. Therefore, scientific studies in the field of medicine have expanded their perspective to new alternatives to counteract viruses such as hepatitis, Ebola, MERS-CoV (Middle East Respiratory Syndrome), H7N9 (avian influenza virus), Crimean-Congo fever, among others [310].

The bioactive compounds present in many foods have been of great interest for their antiviral capacities, reducing infected cells’ activity and inhibiting multiplication in the body. Flavonoid compounds such as tangeretin, nobiletin, and hesperidin have shown application as antivirals (Table 18) [311]. Tang et al. [312] obtained tangeretin from citrus fruit peels and demonstrated a high capacity to inhibit the Lassa virus’s entry (the principal-agent causing viral hemorrhagic fever, VHF) to the host cells and blocking the viral fusion.

Currently, the world is affected by the pandemic crisis caused by the severe acute respiratory syndrome virus (SARS-CoV-2), a descendant of the SARS-CoV coronavirus family, which causes the coronavirus disease (COVID-19) [317]. This virus consists of a genome of positive-sense single-stranded RNA, enveloped by a lipid membrane with structural proteins. The structural proteins are the protein S (glycoprotein homotrimer) responsible for the appearance and recognition of receptors on the target cell, the protein M (membrane glycoprotein) with greater abundance on the surface of the virus, which defines the shape of the lipid membrane, and protein E (a small membrane protein) involved in several processes of the viral cycle [318]. Due to the accelerated increase in cases of contagion and the chronic affectations that this virus presents, alternative drugs, and compounds have been studied for the prevention and inhibition of the virus, such as Arbidol, with a direct antiviral effect on the early viral replication of the virus, Remdesivir, which has a significant effect against a wide range of RNA virus infections, and Lopinavir, a protease inhibitor [18]. Because it is a recently discovered virus, studies in the formulation of a vaccine require an intense investigation of highly efficient compounds to eliminate the virus [318]. However, medical advances in antibody development have accelerated vaccine development. At least 240 vaccines are currently in process (phase I), 44 in clinical trials (phase II) at the time of the writing of this review, so the number of vaccines in development increases daily [319]. Six vaccines are currently in phase III [Johnson & Johnson (United States), AstraZeneca (University of Oxford, United Kingdom), Sinovac (China), Sinopharm (China), Moderna (United States), and Pfizer (United States)] [320]. In this phase, the vaccine’s safety and efficacy against the SARS-CoV-2 virus are more fully evaluated until reaching the final phase (IV), where it is approved [320]. According to the World Health Organization (WHO) report, vaccines’ effect in providing an immune barrier still needs more research [321]. The food loss and wastes present bioactive compounds that generate medicinal effects such as the inhibition of viral activity and the reduction of inflammation in the affected organism, such as hesperidin, a phenolic compound with antioxidant, anti-inflammatory properties, which production could be less expensive. For this reason, the potential of hesperidin, mostly found in the by-products of citrus fruit peels, has been studied (Table 19).

## 6. Conclusions and Perspectives

The agri-food industry generates a large amount of waste that can be applied for food, medicinal, and skincare products by extracting bioactive compounds. Flours obtained from seeds and skins of fruits and cereals presented better sensory and nutritional characteristics in the partial substitution of 10, 20 and 30% of wheat flour and corn in the preparation of bread and cookies, having a consumer acceptance of 90 to 100%. Fruit peels are a good source of colorants that, in addition to contributing to the color in bakery products, provide antioxidant and anti-inflammatory capabilities, proposing their application in the development of new functional foods. Likewise, they are an excellent substrate to produce bromelain enzymes of great interest at an industrial level. Studies carried out to date indicate that the phenolic compounds present in citrus fruit residues have a broad medicinal activity, increasing the protection capacity against chronic diseases such as cancer thanks to their antioxidant, anti-inflammatory, antibacterial properties. On the other hand, tangeretin, nobiletin, and hesperidin turn out antigens’ inhibitors fighting respiratory and hepatic viruses, exhibiting a potential application in pharmaceutical applications. To develop the industry in the extraction of natural colorants from FLW, extraction methods must be carried out more thoroughly on a pilot scale, expanding the panorama and the techno-economic feasibility due to consumers’ great demand to acquire clean product labels.

The reviewed studies demonstrated the potential use of food loss and waste, and its future application on an industrial scale, with lower energy consumption processes. The studies also highlight the potential of the by-products for the human and animal feed population, avoiding excessive consumption of raw materials. There is little literature on the study and analysis of colorants extracted from food losses and waste applied to confectionery products. It is essential to investigate the possible application of naturally derived naturals colorants in beverages, sweets, and candies in the confectionery industry. It is essential to investigate the potential application of derived natural dyes in beverages, sweets, and candies in the confectionery industry. Similarly, the appropriate doses should be thoroughly investigated to effectively counteract viral diseases with phenolic compounds derived from fruit peels, generating new alternatives for low-cost medicine production.

## Figures and Tables

**Figure 1 molecules-26-00515-f001:**
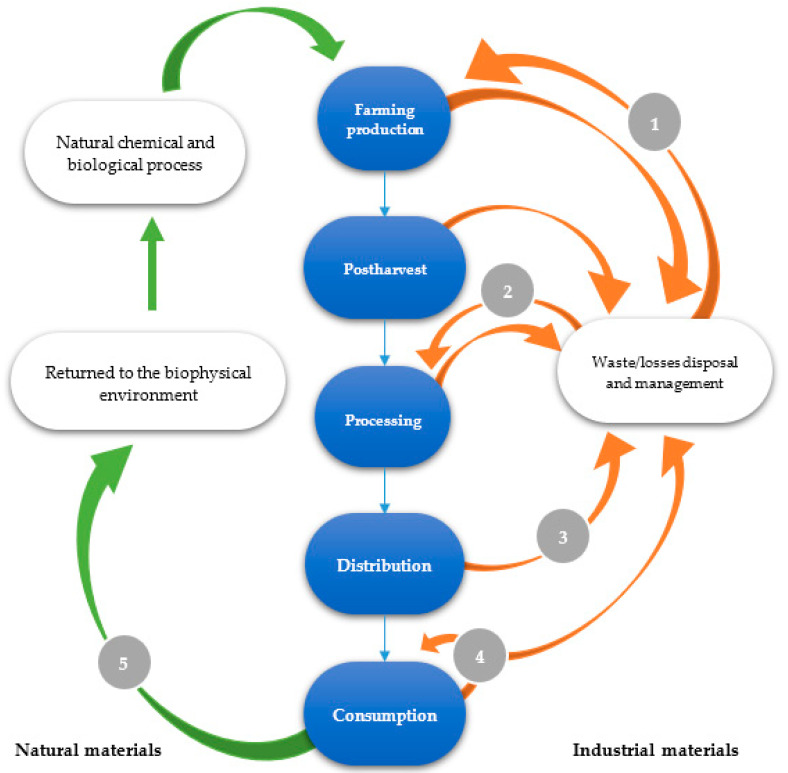
General scheme of waste and losses of the food chain in the circular economy: (**1**) Raw material recovery. (**2**) Modification and transformation of losses. (**3**) Reincorporation of food waste and losses. (**4**) Reuse of packaging. (**5**) Reincorporated food waste [38,39].

**Figure 2 molecules-26-00515-f002:**
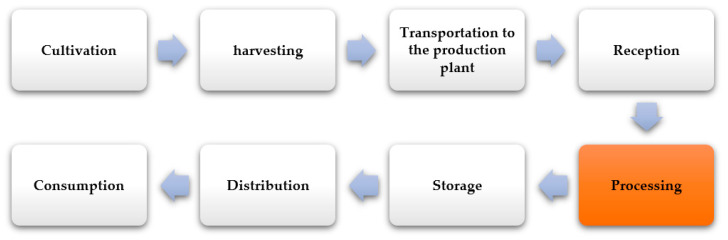
Scheme of the productive chain of the agri-food industry. Source: Adapted from [76].

**Figure 3 molecules-26-00515-f003:**
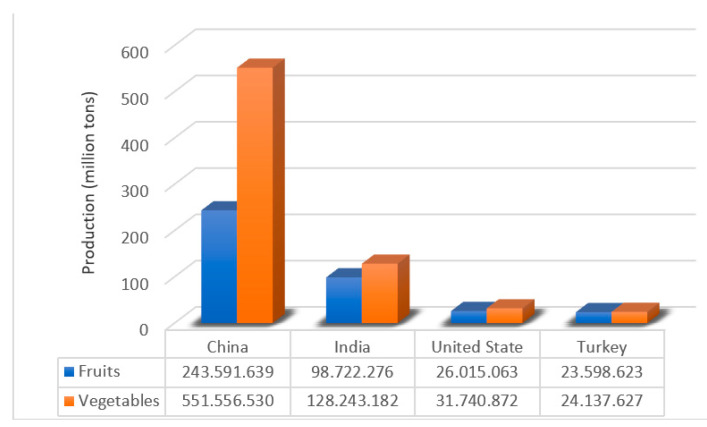
Primary fruit and vegetable producer (2018 production) [81].

**Figure 4 molecules-26-00515-f004:**
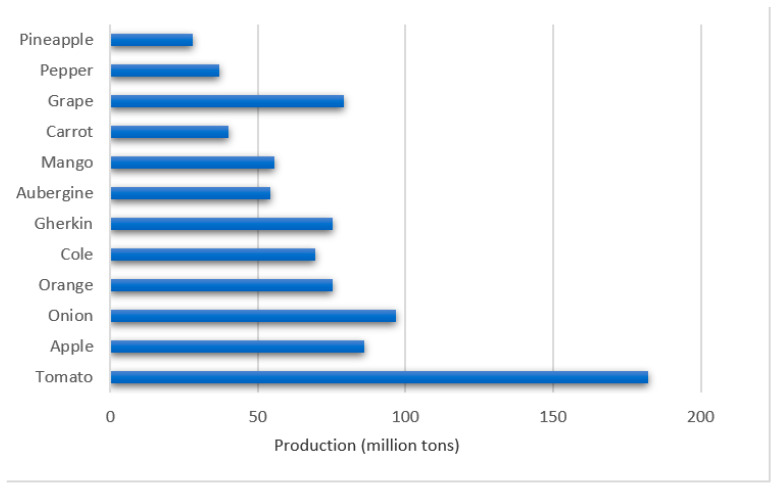
Fruits and vegetables with the highest production worldwide [81].

**Figure 5 molecules-26-00515-f005:**
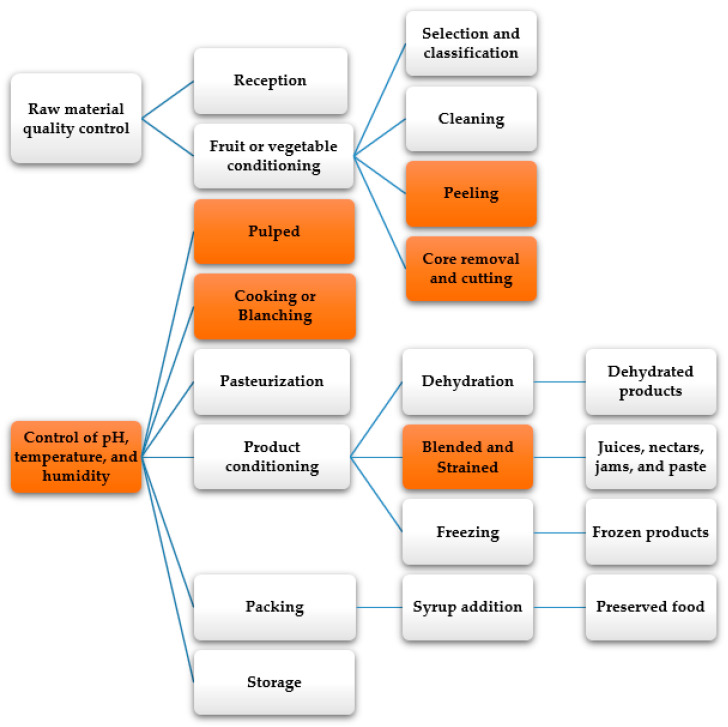
General diagram of fruit and vegetable industrial processing [89,90].

**Figure 6 molecules-26-00515-f006:**
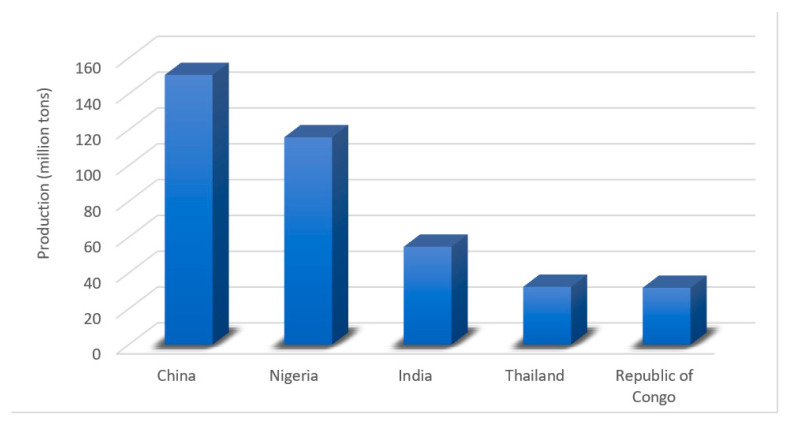
The five leading producer countries of roots and tubers in 2018 [81].

**Figure 7 molecules-26-00515-f007:**
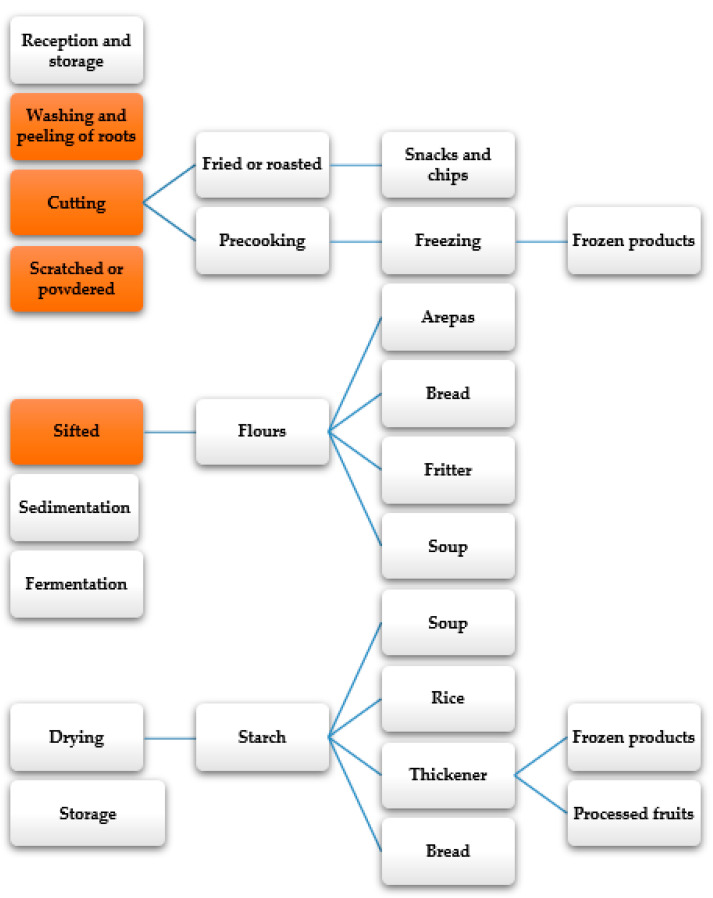
General scheme of root and tuber processing [143,144].

**Figure 8 molecules-26-00515-f008:**
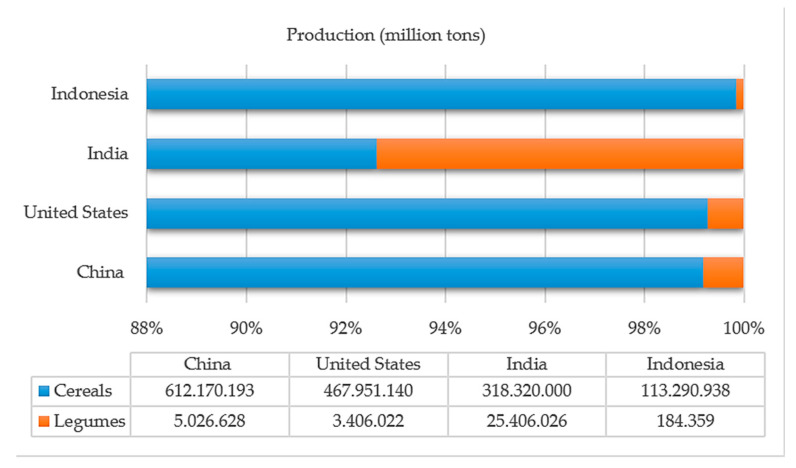
Leading countries in the production of cereals and legumes in 2018 [81].

**Figure 9 molecules-26-00515-f009:**
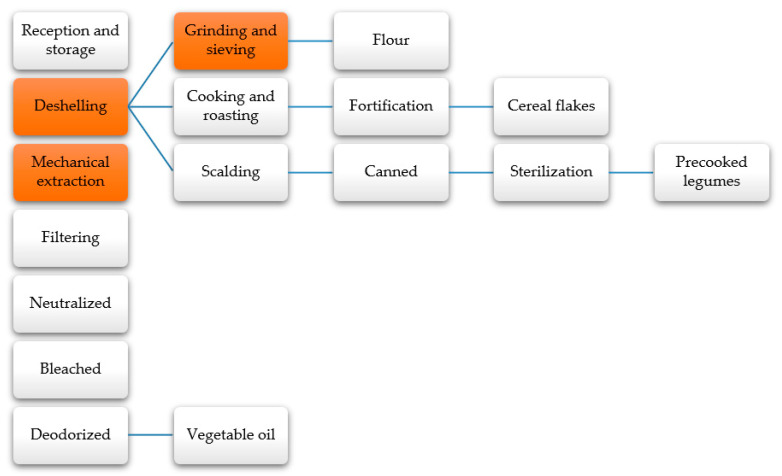
Industrial processing of cereals and legumes [181,182].

**Figure 10 molecules-26-00515-f010:**
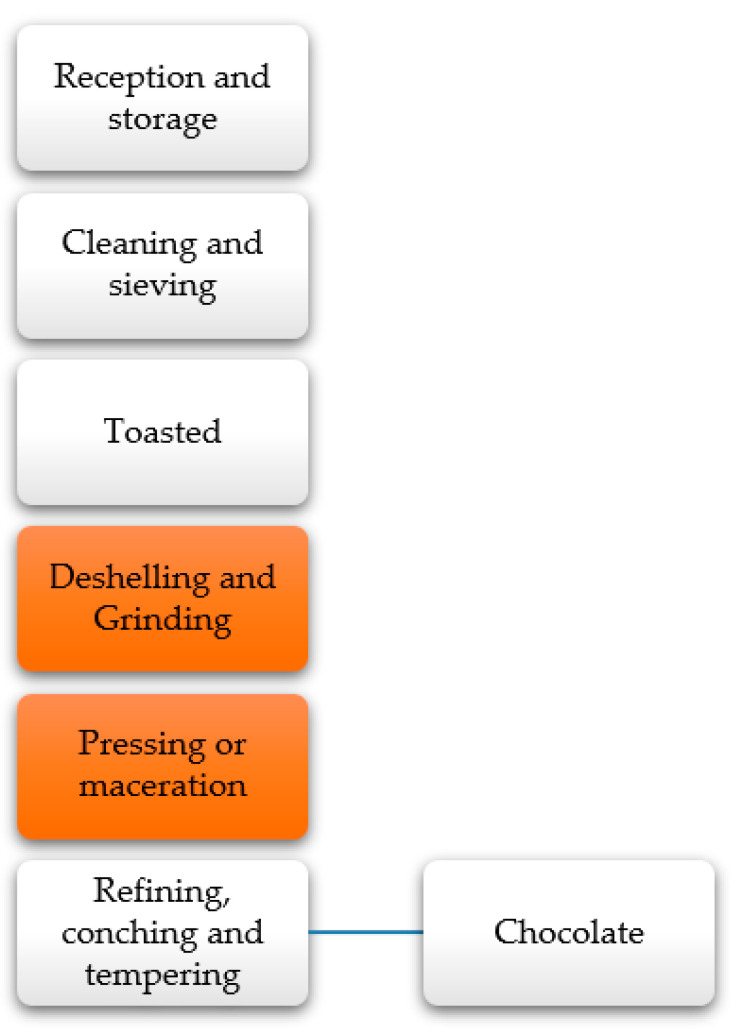
General diagram of cocoa processing [187].

**Figure 11 molecules-26-00515-f011:**
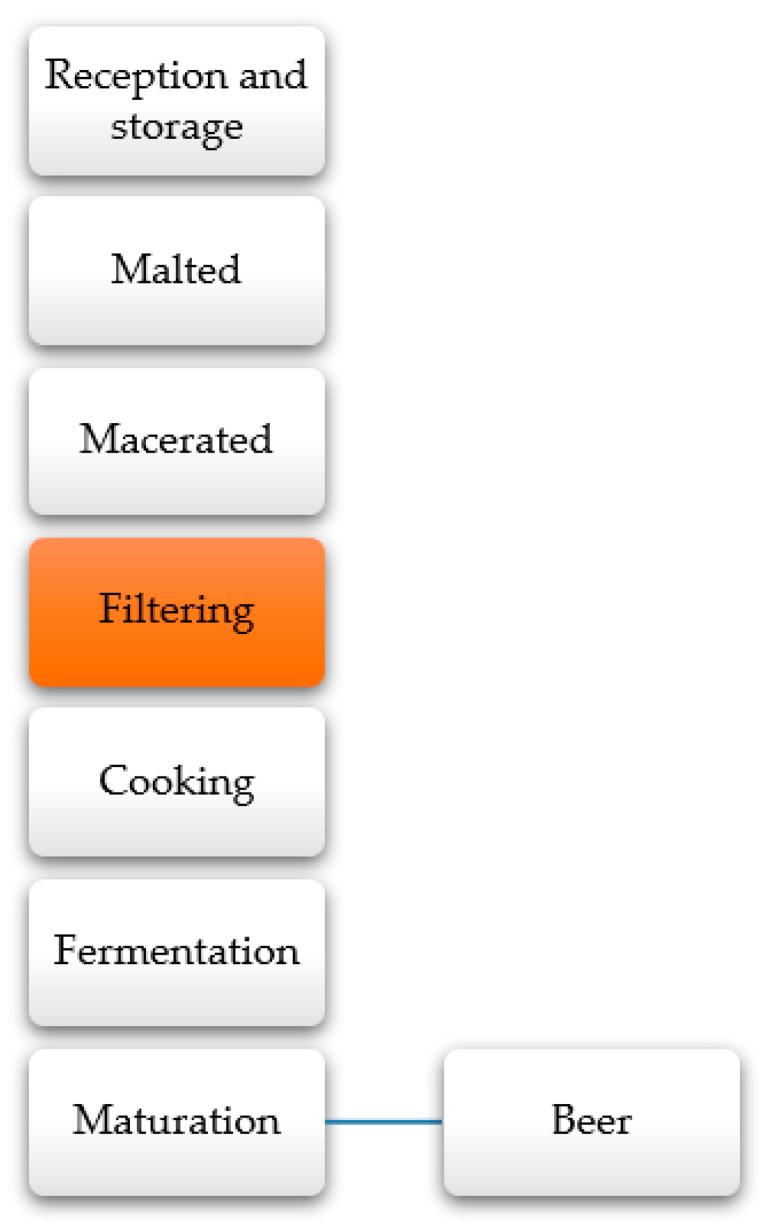
Industrial beer processing scheme [190].

**Table 1 molecules-26-00515-t001:** Technologies used in several countries around the world for the re-use of food waste.

Technology	Countries	Advantage	Disadvantage	Product	References
Anaerobic digestion	United States, Germany, Switzerland, Italy, Spain, and Norway	An utterly harmless end product rich in nutrients is obtained and can be used in agriculture. Solid matter is reduced in the digestion. The gas generated in the stabilization of solids can be used as an energy source. As the tanks are closed, there are no terrible odors outside the premises. During the stabilization process, pathogens and individual parasitic organisms are eliminated	Requires high amounts of water. It is a slow process that requires more time. Requires more energy in processes	Biogas and liquid effluent used as compost	[60,63,64,65]
Composting and vermicomposting	Germany, Switzerland, Italy, Spain, Norway, South Korea, Japan, and several Latin American countries	Reduces the use of inorganic fertilizers. Saves irrigation water due to the water holding capacity of the compost. Provides the necessary nutrients for the development of plants naturally	In the more advanced industrial processes, the use of energy can be considerable, presenting high costs	Fertilizer and compost for crops	[60,63,65]
WTE-Waste to Energy	Germany, Switzerland, Italy, Spain, Norway, South Korea, and Japan	Avoid methane emissions caused by landfills. They offset greenhouse gas emissions caused by the production of electricity from fossil fuels. Recover/recycle valuable resources	It is an expensive process. It pollutes the environment due to incinerators that produce smoke during the combustion process	Energy	[60,66,67]
MBT- Mechanical Biological Treatment	Germany, Spain, Switzerland, Korea, and Japan	Confined material is inert. Conservation of resources and reduction of emissions harmful to the environment	Potential for odor problems. A variety of occupational health and safety problems	A stabilized organic fraction, recovered combustible solid products, materials ferrous/non-ferrous and biogas	[60,68,69]
Feed production	Germany, France, Italy, Korea, and Japan	Reduce pollution and environmental impact that these products generate when they are thrown away as waste. Possibility of reducing production costs in specific diets. Provide the animal with a rich source of protein and energy	Seasonality and variability of production There must be a previous study for the compatibility between the animal and the feed. There must be a regulation for detailed control of food, traceability, and food safety	Animal feeding	[60,63,65]

**Table 2 molecules-26-00515-t002:** Examples of Food Losses and Waste (FLW) generated from the processing of fruits and vegetables worldwide.

Industrial Product	Process	By-Product	Fraction of Raw Material in Fresh Weight (%)	References
Canned pineapple	Peeling	Peel	44	[93]
	Core remove	Core	15	
Canned peaches	Peeling	Peel	8.6	[94]
	Cutting	Seed	11.4	
Dried apple	Core remove	Core	2–4	[92,95]
Tomato paste	Peeling	Peel	3	[95,96]
	Pulping	Seed	7	
	Straining	Pomace	10–20	
Apple juice	Peeling	Peel	5	[95,96]
	Pulping	Seed	2–4	
	Straining	Pomace	20–30	
Pineapple juice	Core remove	Core	11	[95]
	Peeling	Peel	40	
	Straining	Pomace	8	
Mango juice	Peeling	Peel	12–15	[96]
	Pulping	Seed	15–20	
	Straining	Pomace	5–10	
Peach juice	Pulping	Seed	4–11	[96]
Grapefruit juice	Peeling	Peel	30	[96,97]
	Pulping	Seed	20	
Grape juice	Straining	Pomace	13	[92,98]
	Pulping	Seed	5	
Orange juice	peeling	peel	30–40	[96,99]
	Pulping	Seed	2	
	Straining	Pomace	28	
Carrot juice	Straining	Pomace	30–40	[95]
Passion fruit juice	Peeling	Peel	40	[96]
	Pulping	Seed	10	
Pineapple jams	Peeling	Peel	44	[93,100]
	Core remove	Core	11	
Pepper jams	Core remove	Seed	50–60	[99]
Frozen mango	Peeling	Peel	12–15	[88,96]
	Pulping	Seed	13.5	
	Cutting	Discarded pulp	5–10	
Frozen melon	Peeling	Peel	8	[100]
	Cutting	Seed	5	

**Table 3 molecules-26-00515-t003:** Chemical composition of fruit peels.

Peels	Protein Crude (g/100 g)	Fiber Crude (g/100 g)	Ash (g/100 g)	Lipid (g/100 g)	Carbohydrate (g/100 g)	Reference
Pineapple (f.w.)	5.11	14.80	4.39	5.31	55.52	[104]
Tomato (d.w.)	11.17	76.13	3.54	4.48	-	[105]
Apple (f. w.)	2.80	13.95	1.39	9.96	59.96	[104]
Mango (f.w.)	5.00	15.43	3.24	4.72	63.80	[104]
Grapefruit (d.w.)	4.22	1.61	2.99	2.01	46.44	[106]
Orange (f.w.)	9.73	14.19	5.17	8.70	53.27	[104]
Passion fruit (d.w.)	4.05	19.20	7.52	0.10	21.28	[107]
Melon (f.w.)	9.07	29.59	11.09	1.58	48.67	[108]

f.w. = fresh weight; d.w. = dry weight.

**Table 4 molecules-26-00515-t004:** Chemical composition of fruit and vegetable seeds.

Seeds	Protein Crude (g/100 g)	Fiber (g/100 g)	Ash (g/100 g)	Lipid (g/100 g)	Reference
Tomato (d.w.)	24.5	33.9	3.0	20.0	[112]
Peach (d.w.)	27.85	60.92	3.41	1.21	[113]
Apple (d.w.)	34.00	24.00	4.1	27.7	[114]
Grape (f.w.)	6.93	38.60	4.50	2.87	[115]
Mango (f.w.)	7.70	10.23	2.26	65.46	[116]
Melon (d.w.)	27.41	25.32	4.83	30.65	[117]
Pepper (d.w.)	21.29	38.76	4.94	23.65	[118]
Orange (d.w.)	3.06	5.50	2.50	54.20	[119]
Passion fruit (f. w)	15.00	41.33	1.40	29.60	[120]

f.w. = fresh weight; d.w. = dry weight.

**Table 5 molecules-26-00515-t005:** Chemical composition of the pomace and core fruit.

Fruit	Protein (g/100 g)	Fiber (g/100 g)	Ash (g/100 g)	Lipid (g/100 g)	Reference
Grape pomace (d.w.)	8.49	46.17	4.65	8.16	[123]
Apple pomace (f.w.)	3.30	42.10	1.10	2.30	[128]
Mango pomace (d.w.)	10.59	6.14	2.76	1.74	[129]
Tomato pomace (d.w.)	19.27	59.03	3.92	5.85	[130]
Peach pomace (d.w.)	7.50	54.20	3.00	3.00	[131]
Pineapple core (f.w.)	0.85	47.60	1.30	3.17	[132]
Carrot pomace (d.w.)	20.90	55.80	5.50	1.30	[133]

f.w. = fresh weight; d.w. = dry weight.

**Table 6 molecules-26-00515-t006:** Food losses and waste (FLW) generated from the global processing of roots and tubers.

Industrial Product	Process	By-Product	Fraction of Raw Material in Fresh Weight (%)	References
Frozen potato	Peeling	peel	6–10	[146,147]
Frozen yucca	Peeling	Peel	3–5	[148,149]
Cassava starch	Peeling	Peel	3–5	[150]
	Sifted	Bagasse	10	
Potato starch	Peeling	Peel	10–15	[97]
Yam starch	Peeling	Peel	4–5	[97]
Cassava flour	Peeling	Peel	3–5	[151]
	Sifted	Bagasse	6	
Potato chips and snacks	Peeling	Peel	15–20	[152,153]
Cassava chips and snacks	Peeling	Peel	7–10	[65]

**Table 7 molecules-26-00515-t007:** Chemical composition of root and tuber peels.

Peels	Cellulose (%)	Hemicellulose (%)	Lignin (%)	Ash (%)	Protein (%)	Lipid (%)	References
Cassava (f.w.)	37.90	23.90	7.50	4.50	4.73	0.19	[158,159]
Yam (f.w.)	-	-	-	4.45	6.60	1.12	[160]
Potato (f.w.)	15.00	39.00	10.00	6.34	8.00	2.60	[161,162]

f.w. = fresh weight; d.w. = dry weight.

**Table 8 molecules-26-00515-t008:** Bagasse composition from cassava processing.

Bagasse	Cellulose (g/100 g)	Hemicellulose (g/100 g)	Lignin (g/100 g)	Ash (g/100 g)	Protein (g/100 g)	Lipid (g/100 g)	Reference
Cassava	21.47	12.97	21.86	12.60	13.31	1.18	[165]

**Table 9 molecules-26-00515-t009:** Food Losses and Waste (FLW) generated from the processing of cereals and legumes.

Industrial Products	Process	By-Product	Raw Material Fraction (%)	Reference
Chocolate (f.w.)	Deshelling	Shell	10–12	[196]
Soy oil (d.w.)	Deshelling	Shell	8	[197]
	Extraction	Cake	60	
Beer (d.w.)	Filtering	Bagasse	20–25	[192]
Wheat flour (f.w.)	Deshelling	Shell	15–20	[198]
Corn flour (d.w.)	Deshelling	Shell	6	[199]
Rice flour (f.w.)	Deshelling	Shell	28	[200]
Canned beans (f.w.)	Deshelling	Shell	6–8	[201]
Canned peas (f.w.)	Deshelling	Shell	10	[201]

f.w. = fresh weight; d.w. = dry weight.

**Table 10 molecules-26-00515-t010:** Chemical composition of cereal and legume husks.

Husk	Cellulose (%)	Hemicellulose (%)	Lignin (%)	Ash (%)	Protein (%)	Lipid (%)	References
Cocoa (d.w.)	35.40	37.00	14.70	12.30	16.70	36.9	[158,202]
Corn (d.w.)	31.00	34.00	14.30	6.8	-	-	[203]
Wheat (d.w.)	36.00	18.00	16.00	-	6.00	5.00	[204]
Soybean (d.w.)	5.10	19.14	2.10	13.85	23.80	47.00	[205,206]
Rice (d.w.)	40.00	21.00	22.00	16.00	-	-	[207]
Beans d.w.)	50.25	1.23	-	3.51	-	-	[208]
Pea (d.w.)	17.00	33.00	2.50	-	14.30	-	[209]
Oat (d.w.)	17.20	33.10	25.40	6.00	1.70	0.50	[210]

f.w. = fresh weight; d.w. = dry weight.

**Table 11 molecules-26-00515-t011:** Composition of malt bagasse in beer processing.

Bagasse	Fiber (%)	Carbohydrate (%)	Protein (%)	Lipid (%)	Ash (%)	Reference
Malt (dry weight)	63.84	73.84	23.60	4.44	2.78	[211]

**Table 12 molecules-26-00515-t012:** Composition of the waste cake from soybean oil processing.

Cake	Protein (%)	Fiber (%)	Ash (%)	Calcium (%)	Phosphorus (%)	Reference
Soybean (dry weight)	47.50	5.10	6.40	0.13	0.64	[197]

**Table 13 molecules-26-00515-t013:** Applications of flours obtained from agri-food by-products in functional foods preparation.

By-Product	Application	Added Amount (%)	Effect of By-Product Flours on Functional Foods	Reference
Seed, pomace, and grape peel	Baked goods and pasta	2, 4, and 6 (seeds); 3, 6, and 9 (pomace); 2, 5, and 10 (pomace), respectively	Increase of functional ingredients in the bakery, pastry, and pasta industries without damaging the quality of the products	[225]
Pigeon pea cotyledon	Biscuit	95, 90, and 85 of wheat flour; 5, 10, and 15 pigeon pea flour	The protein content of the cookies enriched with pigeon pea cotyledon 1.4 times and a significant increase in fiber content	[226]
Grape peel, seeds, and remains of the pulp	Biscuit	30 and 40 replacement with wheat flour	The flour presented physicochemical characteristics within the nutritional standards, with an acceptance of 92.6% of the consumers evaluated	[227]
Grape pomace	Biscuit	10, 20, 30, 40, and 50 replacement of grape pomace flour with wheat flour	The protein, fiber, and ash content increased considerably in the product, with consumer acceptability of 100%	[228]
Pomegranate, grape, and rosehip seeds	Turkish noodles	10, 20, and 30 inclusion of grape pomegranate and rosehip, respectively	The addition of 10% of grape, pomegranate, and rosehip seeds increased the antioxidant activity of the product between 5.7 and 8.4 times	[229]
Apple pomace and sugarcane bagasse	Corn extrudates (high fiber croquettes)	15–30 cornmeal substitution	The inclusion of apple pomace and sugarcane bagasse showed the potential to produce extrudates with significant expansion, with relatively lower energy inputs and high fiber content	[230]
Olive pomace	Oat and rice extrudates	5 and 10 substitutions of rice flour and oats	Favorable impact on the physical characteristics of the extruded product for the base of rice flour and oats. High content of fiber, protein, and polyphenols through the incorporation of olive pomace	[231]
Apple pomace	Sorghum and corn extrudates	10, 20, and 30 substitutions with sorghum and cornflour	Extruded products with better phenolic content, antioxidant activity, textural, and functional properties with greater inclusion of apple pomace were obtained	[232]
Pineapple peels	Cereal bars	3, 6, and 9 inclusion of pineapple peel flour	Adding up to 6% pineapple peel flour to cereal bars did not alter sensory acceptance and increased fiber content	[233]
Grape seeds	Cereals, pancakes, and noodles	Substitution of 25 and 30 of grape seed flour in pancakes, 20 in noodles, and 5 in cereal bars	The cereal bars, pancakes, and noodles had a high acceptance by the consumer	[234]
Watermelon seeds	Biscuit	Substitution with 5–10 of watermelon seed flour in Biscuits	Watermelon in concentrations of 5–7.5% in cookies improves quality and increases the protein content	[235]

**Table 14 molecules-26-00515-t014:** Studies carried out in the extraction of colorants from agri-food by-products.

Pigment	By-Product	Extraction Method	Production Scale	Applications in Food Products	Reference
Cyanidin	*Ficus carica L.* peel	Heat, microwave, and ultrasound	Laboratory scale	Additive and natural coloring in pastry and confectionery	[242]
Cyanidin 3-glucoside	Coffee exocarp	Solvent Extraction (SC) and Solid Solvent (SSR)	Laboratory scale	Natural coloring applied to French meringue	[243]
Cyanidin 3-glucoside	The by-product of the mulberry industry	By-product mixture with water 1: 3 (*w*/*v*) stirred for eight hours in the dark, filtered through a sieve, and evaporated to 1/3 of the initial volume	Laboratory scale	Additive and natural coloring	[244]
Quercetin, Cyanidin 3-O- glucoside	Solid waste and onion leaves	Extraction by ecological solvents, such as water and glycerol. 2-Hydroxypropyl-β-cyclodextrin was also used as a co-solvent to increase the extraction yield.	Pilot-scale	Natural coloring applied to yogurts, providing antioxidant properties	[245]
β-carotene	Tomato peel and seed	Ultrasound extraction and application in biofilms	Laboratory scale	Edible biofilm coloring	[246]
Bethany and iso-betanin	Peel and waste pulp of red prickly pear	Pulsed electric field and ultrasound	Pilot-scale	Natural dyes. Protection against low-density lipoprotein (LDL) oxidative modifications	[240]
Lycopene, phytoene, phytofluene, and β-carotene	Tomato peel	Extraction using organic solvents (ethanol and acetone)	Laboratory scale	Natural colors in sunflower oil and spaghetti	[247]
β-carotene, γ-carotene, and δ-carotene	*Astrocaryum vulgare* and *Bactris gasipaes* peel	Extraction using organic solvents (ethanol and acetone)	Laboratory scale	Natural coloring, in addition to being a source of vitamin A	[248]
Lutein and zeaxanthin	Orange peels	Use of the fungi *Monascus purpureus* and *Penicillium purpurogenum* by fermentation in solid-state (SSF) and semi-solid (ST) and aqueous solutions of ethanol 70% *v/v* and isopropanol 95% *v*/*v*	Laboratory scale	Natural coloring in pastry	[249]
Lycopene, phytoene, phytofluene, β-carotene, cis-lycopene, and lutein	Tomato peel	Dried at 40 °C, ground and passed through 0.15 mm, using ethanol as solvent and acetone/hexane (50:50 *v*/*v*). The method described by Hackett et al. [250]	Laboratory scale	Natural coloring and antioxidant in ice cream	[251]

**Table 15 molecules-26-00515-t015:** Extraction of enzymes from agri-food by-products.

Enzyme	By-Product	Method	Result	References
Bromelain	Pineapple peel	An aqueous biphasic system consisting of polyethylene glycol (PEG) and polyacrylic acid	Bromelain extraction achieved a yield of 335.27% with a purification factor of 25.78 in PEG	[256]
	Pineapple stems	Aqueous two-phase micellar systems with ionic liquids as co-surfactants	A high bromelain recovery (90%) was found for the low micelle phase for all systems	[258]
	Pineapple peel	Gemini surfactant-based reverse micelle	Activity recovery, purification fold, and protein extraction efficiency were 160 to 163%, 2.7 to 3.3, and 59%, respectively	[259,260]
	Pineapple stems and skins	Precipitation with carrageenan using a factorial experimental design	High recovery yield 80–90% of active bromelain for both (stems and peels), making it possible to obtain approximately 0.3 g of bromelain from 100 g of pineapple by-products	[261]
	Peel, pulp residues, and pineapple core	Extraction with ethanolic polyphenol from cashew leaves	Solids were obtained from pineapple waste with the following protease activities: 5343 CDU/g (core), 5147 CDU/g (shell), and 5732 CDU/g (pulp)	[124]
Lipases	Bagasse and shell of Pear *var*. Hamlin, Valencia, and Natal	Emulsified olive oil and on p-nitrophenyl substrates	The bagasse and peel lipases from different orange varieties were obtained at an optimum pH between 6.0–8.0 and an optimum range temperature between 30 to 60 °C	[262]
	Industrial waste from palm oil	Use of selected strain of *Aspergillus niger* grown in solid-state (SSF) and submerged fermentation (SMF)	Lipase activity levels of up to 15.41 UI/mL were reached in the palm oil extract by SSF	[263]
	Olive oil cake	Use of *Candida utilis* in solid-state fermentation	An alkaline substrate treatment appeared to be effective, leading to 39% increases in lipase production	[264]
	Mango seeds and peel	Use of *Yarrowia lipolytica* in submerged fermentation	The highest lipase activity observed was 3500 UI/L of the solid residue	[265]
Amylases	Mango seeds	Use of *Aspergillus niger* isolated	The maximum α-amylase activity was obtained	[266]

**Table 16 molecules-26-00515-t016:** Agri-food by-products as a photoprotective source in the formulation of sunscreens.

By-Product	Study	Result	Reference
Passion fruit seed	Formulation of sunscreen as foundation and correction makeup	The 3% passion fruit polar extract correctors had a sun protection factor (SPF) of 18.09 ± 1.48 and 18.60 ± 1.21. Did not cause skin irritation when evaluated in human volunteers	[275]
Grapeseed	Evaluation of the protective effects of grape seed on fibroblasts irradiated with UV light	Grape seed increased cell viability and effectively protected fibroblasts from UV damage, also improving UV filter absorbency and overall formulation efficacy	[267]
Oat shell and walnut	Determination of functional properties such as antioxidant power and UV absorption capacity of lignans extracted from by-products	The extracted lignans showed adequate protection against UV radiation, a property of great interest to block the entire ultraviolet spectrum	[276]
Industrial coffee waste	Biological effects of the by-product in the development of a new generation of solar filters	The emulsion containing 35% (*w*/*w*) of the coffee fraction showed improvements in water performance with a broad spectrum of sun protection (SPF) compared to an emulsion containing 35% (*w*/*w*) of green coffee that improved the SPF in physical sunscreens	[277]
Waste from the guava industry	Formulation of a cosmetic product as a sunscreen	The polar extract showed synergy with the UV chemical filter (Ethylhexyl methoxycinnamate), enhancing the sun protection factor by 17.99%	[278]
Grape seeds	By-product effect and its UV and visible light protection capacity	The by-product showed the ability to absorb a wide range of the solar spectrum, including ultraviolet and blue light. At a concentration of 100 µg/mL, it significantly inhibited nitrous oxide production in RAW 264.7 cells (commonly used model of mouse macrophages for the study of cellular responses)	[279]
Rambutan skin	Develop a sunscreen based on rambutan skin polar extracts	The by-product can synergistically increase the sun protection factor (SPF) values of synthetic organic sunscreens and lower costs in a sunscreen formulation	[280]
Shells and seeds Amazonian fruits	Determine the protection capacity of polar seed extracts against UV light	The polar extracts showed high capacity, especially *Caryocar villosum*, *Garcinia madruno*, and *Bertholletia excelsa* with UV absorption peaks, and piquiá, bacurizinho, and açaí seeds (*Euterpe oleracea and E. precatoria*)	[281]

**Table 17 molecules-26-00515-t017:** Studies for the efficiency of fruit and vegetable by-products as antibacterial, anti-inflammatory, and anticancer agents.

By-Product	Study	Result	Reference
Orange peel (*Citrus sinensis*)	Phenolic extracts of the orange peel against oxidative stress in primary human colon tumor (Caco-2) and metastatic cell lines (LoVo and LoVo/ADR)	Orange peels showed antioxidant properties by inhibiting the activities of total matrix metalloproteinases (MMP), preventing the progression of cancer and the tumor cell metastasizing	[289]
Orange peel (*Citrus sinensis*)	Matrix metalloproteinase (MMP) and proteasome activities in cells by peel extracts for colon cancer	The extracts inhibited proteasome (responsible for the degradation of cell regulatory proteins, found in excess in tumor cells) activity in extract-treated cells, inhibiting the progression of cancer	[299]
Peel of *Citrus aurantifolia* and *Citrus reticulata*	Citrus peel like a natural model for cancer prevention and treatment	Hesperidin and naringenin increased the cytotoxicity of doxorubicin in cancer cells (MCF-7 and HeLa). Hesperidin can lower the concentration of liver and serum lipids and reduce osteoporosis	[300]
Pineapple peel (*Ananas comosus*)	Determination of extraction methods, purification, and therapeutic applications of bromelain	Medicinal applications: inhibition of platelet aggregation, antithrombotic, anti-inflammatory, modulation of cytokines, and immunity	[301]
Mango seed (*Mangifera indica*)	Cytotoxic effect of the by-product extract on MDA-MB-231 and MCF-7 cancer cells	The extract induced cytotoxicity in MDA-MB-231 and MCF-7 cells with IC50 values of 30 and 15 μg/mL, respectively	[302]
Broccoli residues (*Brassica oleracea var. Italica*)	Inhibitory effect of broccoli sprout extracts on prostate cancer cell phones with low (AT-2) and high (MAT-LyLu) metastatic potential	Efficiency in the inhibition of cell proliferation, motility, and cell coupling was obtained using the extracts	[303]
Grape pomace (*Vitis vinifera*)	By-product effects on metabolism and redox status in HepG2 human hepatocarcinoma cells	Long-term metabolic effects generated cytotoxicity, and cells died from necrosis, and it was not toxic to non-cancerous human fibroblasts	[304]
Grapefruit peel (*Citrus paradisi)*	Grapefruit peel essential oil as an active component against strains of *Candida albicans* isolated from patients with sub-protein stomatitis	*Citrus paradisi* essential oil at 25% and 12.5% presented a limited antifungal activity, and that presents inhibition halos of greater diameter	[290]
Peel and leaves of kafir lime (*Citrus hystrix*)	Inhibitory effect of essential oils of by-products against respiratory pathogens (*Streptococcus pneumoniae*, *Haemophilus influenzae*, *Staphylococcus aureus*, and *Acinetobacter baumannii*) and evaluation of their active components	Twenty-five components identified in the extract, limonene as the main compound with yields of 91.5 and 88.6% of antibacterial activity	[305]

**Table 18 molecules-26-00515-t018:** Different residues as a source of phenolic compounds and their application in the prevention of viral infections.

By-Product	Compounds	Virus	Result	Reference
*Citrus reticulate* peel	Tangeretin	Human respiratory syncytial virus (RSV)	Inhibition of RSV replication and decreased inflammation in lungs	[313]
Peels of *Citrus madurensis Lour*	Hesperidin, diosmin, neohesperidin, nobiletin, tangeretin	Hepatitis B virus	Significantly inhibited the surface antigen of the hepatitis B virus	[314]
Citrus fruit peel	Nobiletin	Hepatitis B virus	Inhibited surface antigen and hepatitis B virus	[315]
*Citrus reticulate* peel	Nobiletin	Chikungunya virus	Reduced the load of the virus and exerts a positive effect on the immune system	[316]

**Table 19 molecules-26-00515-t019:** Research on the efficiency of hesperidin from citrus fruit peels to counteract the SARS-CoV-2 virus.

By-Product	Study	Result	Reference
Peels of lime (*C. Aurantifolia)*, grapefruit (*C. maxima*), lemon *(C. jambhiri*), lime (*C. limetta*), grapefruit (*C. medica L.*), mandarin (*C. reticulata*), and mandarin (*C. reticulate white*)	Identification of the added value of the by-product through different extraction methods and its potential utility against COVID-19	The amount of iron and phenols was higher in grapefruit and grapefruit peel compared to other species, obtaining bioactive compounds and medicinal phytochemicals such as flavonoids for the treatment of COVID-19	[322]
Assorted citrus peels	Review of varieties of citrus by-products as a source of hesperidin, an active compound against COVID-19 and its extraction process	Hesperidin has a strong binding affinity for all significant viruses such as SARS-CoV and their mutations with the potential to prevent it from spreading to cells; likewise, hesperetin (hesperidin aglycone) and naringin stop the pro-inflammatory reaction of the immune system	[288]
Orange peel (*Citrus × sinensis*)	Study of hesperidin as a source to counteract the SARS-CoV-2 virus from citrus fruits	Hesperidin showed a good safety profile, with a median lethal dose (LD_50_) of 4837.5 mg/kg. A dose administration of 500 mg/kg of flavanone did not induce any abnormality in body weight, clinical signs, symptoms, and blood parameters	[18]
Assorted citrus peels	The feasibility of hesperidin for the treatment of COVID-19	Hesperidin targets the root of the infection by binding the protein spike	[323]

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
