# Peer review of "The Potential of Selected Agri-Food Loss and Waste to Contribute to a Circular Economy: Applications in the Food, Cosmetic and Pharmaceutical Industries"

_molecules, 2021, doi:10.3390/molecules26020515_

Round 1

Reviewer 1 Report

Manuscript ID molecules-1055612 is a very important material related to the circular economy in the agri-food sector. The Authors made a very difficult attempt to summarize the possibilities of using waste food products in other sectors of the market. Due to the wide scope of material recycling, making such a summary is probably  impossible. I am thankful for the Authors for taking up the challenge and preparing a review work on this topic. Manuscript may be published, however I suggest you take note of the comments below:

  1. I have doubts whether the manuscript corresponds to the scope and publishing profile of the Molecules journal. I leave this decision in the Editor hands.
  2. The title should be changed because it does not completely cover the topic: The potential of agri-food loss and waste to contribute to a circular economy: Applications in the food, cosmetic, and pharmaceutical industries.
  3. I propose a modification: The potential of selected agri-food loss and waste to contribute to a circular economy: Applications in the food, cosmetic, and pharmaceutical industries.
  4. Please prepare a graphical abstract that will allow for a better understanding of the authors' intentions.
  5. The Circular Economy chapter should present legal regulations related to the use of waste food in a circular economy in various countries around the world.
  6. An interesting chapter would be a review and compilation of popular technologies for the re-use of waste products from the agri-food sector.
  7. The work also does not present the social, economic and environmental aspects of the circular economy in this area.
  8. In the case of using drawings from other publications, the consent of the publisher to which the copyright belongs must be obtained.

Author Response

Reviewer 1

  1. I have doubts whether the manuscript corresponds to the scope and publishing a profile of the Molecules journal. I leave this decision in the Editor's hands.

R// We appreciate the reviewer's comment. However, we are applying to this journal since we were invited to contribute to a particular issue called "Circular Economic: Agri-Food By-products as Source of Bioactive Compounds," We believe the review scope met the requirement of the issue.

  1. The title should be changed because it does not completely cover the topic: The potential of agri-food loss and waste to contribute to a circular economy: Applications in the food, cosmetic, and pharmaceutical industries.

R// We appreciate the reviewer's comment, and the title was modified according to the

reviewer suggestion.

  1. I propose a modification: The potential of selected agri-food loss and waste to contribute to a circular economy: Applications in the food, cosmetic, and pharmaceutical industries.

R// We appreciate the reviewer's comment, and we corrected it according to the proposed title.

  1. Please prepare a graphical abstract that will allow for a better understanding of the authors' intentions.

R// We appreciate the reviewer's comment. A graphical abstract was prepared for a better understanding of the topic.

  1. The Circular Economy chapter should present legal regulations related to the use of waste food in a circular economy in various countries around the world.

R// We appreciate the reviewer's comment. Usually, legal regulations are not easy to reach because several countries have not even implemented them. We summarize some regulations from different countries in Europe and Asia on the use of food waste (were added between lines 155 - 179):

"Based on legal aspects, the analysis or treatment of food waste in a circular economy should focus on the applicable laws of each country [40,41]. Within the legal framework of the European Union (EU) in a global context of regulations and policies for the efficient use of resources and sustainable patterns of consumption and production, it plans to develop advanced routes of different recovery (incorporation into the food industries) to the usual processes (Animal feeding, composting and anaerobic digestion) [42,43]. The re-use and minimization of food waste will be dealt with, provided that these are suitable for human consumption and do not generate toxicity during their treatment [44].

The Treaty on the Functioning of the European Union contemplates the international transport of waste, which under this regulation waste is classified to determine its valuation and re-use subsequently. Agri-food waste (such as peels, seeds, vegetable, and cereal residues) are classified as "green list," which are determined as non-infectious, and it is planned to incorporate them into the treatment process [45,46]. In the transition to a circular economy, the European Parliament initiated the regulation of the product's life cycle in its entirety, from primary production to waste in conjunction with the management and market of secondary raw materials (food by-products) [47,48]. Countries such as Germany, France, and Italy, in the face of established regulations, have promoted government initiatives to use food waste that are not suitable for human consumption in the production of feed and composting [49].

In Japan, the Food Recycling Law was established in 2001 [50], which establishes that companies and industries in the food chain (from agricultural production to consumption) are participants in the re-use of waste and reduced food waste. This law encouraged companies such as Eco-Farm to use plant residues from food industries to make organic fertilizers used in crops that supply the raw material in food production [51]. In countries such as Korea, Taiwan, and Thailand, the use of food waste is promoted and regulated by-laws for its use in feed for ruminants, pigs, and poultry [52]."

  1. An interesting chapter would be a review and compilation of popular technologies for the re-use of waste products from the agri-food sector.

R// We appreciate the reviewer's comment. A chapter of popular technologies for re-using waste from the agri-food sector was added (lines 181 - 200, chapter 3).

  1. The work also does not present the social, economic, and environmental aspects of the circular economy in this area.

R// We appreciate the reviewer's comment. A comment was added where the social, environmental, and economic aspects of the circular economy in the agri-food sector are described (lines 99 - 120).

"The transition towards a circular economy involves various dimensions such as social, economic, and environmental spheres, which generate opportunities for regeneration, renewal, and innovation in the agri-food industries, protecting the scarcity of resources [24]. The integration of higher income from circular activities and the minimization of manufacturing costs would affect demand, supply, and prices, generating indirect effects that accelerate the economy's total growth and varying in positive terms of GDP [25,26]. In the same way, high-quality recycling activities and skilled jobs in the transformation and remanufacturing of food losses and waste creating jobs through the development of local reverse logistics with SMEs generating net savings in raw material costs adopting a circular model [27]. In the environment, the circular economy directly influences the exploitation and deterioration of the ecosystem, reducing the excessive consumption of fertilizers, pesticides, fuel, and non-renewable electricity [28]. Likewise, reducing residues in the food chain by improving productivity and technology is applied in the transformation and regeneration processes of natural materials, making responsible use of soil, aquatic resources, and reducing carbon dioxide emissions [29]. The circular economy's social aspects have a drastic impact on the current culture of food consumption, with lower amounts of waste in homes and greater use of natural and energy resources [30]. In the same way, giving a second utility to packaging by the consumer [31,32]. The problem of social inequality caused by poverty and famine would decrease, obtaining more affordable food and greater availability of jobs due to the new circular technological systems and the creation of companies in conjunction with local industries, forming a fabric that will improve the quality of food products and consumer satisfaction [33,34]."

  1. In the case of using drawings from other publications, the consent of the publisher to which the copyright belongs must be obtained.

R// We appreciate the reviewer's suggestion. The referenced figures of articles and books are of our authorship. However, the tables' industrial processes, quantities, and general information were obtained from the referenced sources. The tables, figures, and diagrams are of our authorship, and it is not required to request permission from publishers as suggested.

Reviewer 2 Report

Paragraph 4.2.1 and table 15:

Authors should describe the type of extracts: are they water extracts or non-polar extracts?

What kind of antioxidants? Phenolic compounds?

Paragraph 4.3:

It is a sensitive topic that highlighted by the authors. However it would be interesting to include a comment about the strategies to obatain by-product extracts for pharmaceutical purposes, including the antimicrobial activity. In this regard, different research groups are moving in order to explore the potential of water and/or hydroalcoholic extracts of byproducts: a green and sustainable approach of circular economy. However, this reduces sometimes the antimicrobial spectrum because the phenolic compounds are the prominent plant secondary metabolites of polar extracts. It is often observed an intriguing antimycotic activity.

Authors' contributions section should be filled according to journal guidelines.

Author Response

Reviewer 2

  1. Paragraph 4.2.1 and table 15: Authors should describe the type of extracts: are they water extracts or non-polar extracts? What kind of antioxidants? Phenolic compounds?

R// We appreciate the reviewer's comment. The type of extract was specified in table 16, and the type of antioxidant used is commented in paragraph 5.2.1 (lines 594 – 601)

"Antioxidants such as phenolic compounds, also defined as polyphenols, are derived from the secondary metabolism of plants [276]. They are commonly found in foods such as vegetables, fruits, and legumes consumed daily by humans, present in the pulp and seeds, skins, and pomace [39]. For instance, extraction and application of phenolic compounds from the loss and waste of food have been of great interest due to their antioxidant capacity and to avoid the massive use of food to preserve food safety worldwide [269]. For that reason, seeds and peels are considered antioxidant sources and are incorporated in the formulation of sunscreens (Table 16)."

  1. Paragraph 4.3: It is a sensitive topic highlighted by the authors. However, it would be interesting to include a comment about the strategies to obtain by-product extracts for pharmaceutical purposes, including the antimicrobial activity. In this regard, different research groups are moving in order to explore the potential of water and/or hydroalcoholic extracts of by-products: a green and sustainable approach of circular economy. However, this reduces sometimes the antimicrobial spectrum because the phenolic compounds are the prominent plant secondary metabolites of polar extracts. It is often observed an intriguing antimycotic activity.

R// We appreciate the reviewer's comment. A discussion was added about the most ecological alternatives for extracting phenolic compounds, their effectiveness at the laboratory scale and semi-industrial scale (line 624 - 634).

"The extraction of these compounds from by-products is carried out by several routes [293]. In the case of phenolic extracts such as hesperidin, nobiletin, and tangeretin, there are currently ecological techniques such as hydroalcoholic extraction [294], which is 90% efficient both on a laboratory scale and on a pilot scale, and hydrodistillation applied mostly to the shells of citrus, which provides 17% more performance than conventional methods [294,295]. In the same way, controlled hydrodynamic cavitation is the route with the highest performance overcoming acoustic cavitation to pilot scale for most applications, including natural products extraction [296,297]. Ultrasonic extraction with organic solvents offers advantages with much faster extraction rates compared to the medium and low energy consumption in conjunction with the MTT assay method that reflects the cellular viability as a result of the antimicrobial capacity of bioactive compounds [298]."

  1. Authors' contributions section should be filled according to journal guidelines.

R// We appreciate the reviewer's comment. We added the contribution of all of the authors according to the journal guidelines.

Round 2

Reviewer 1 Report

Many thanks to Athors for improving manuscript. In my opinion paper can Bu publish in current form. Congratulations !